# Mechanisms of Coronavirus Nsp1-Mediated Control of Host and Viral Gene Expression

**DOI:** 10.3390/cells10020300

**Published:** 2021-02-02

**Authors:** Keisuke Nakagawa, Shinji Makino

**Affiliations:** 1Laboratory of Veterinary Microbiology, Joint Department of Veterinary Medicine, Gifu University, Gifu 501-1193, Japan; nkgwk@gifu-u.ac.jp; 2Department of Microbiology and Immunology, The University of Texas Medical Branch, Galveston, TX 77555-1019, USA; 3Center for Biodefense and Emerging Infectious Diseases, The University of Texas Medical Branch, Galveston, TX 77555-1019, USA; 4UTMB Center for Tropical Diseases, The University of Texas Medical Branch, Galveston, TX 77555-1019, USA; 5Sealy Center for Vaccine Development, The University of Texas Medical Branch, Galveston, TX 77555-1019, USA; 6Institute for Human Infections and Immunity, The University of Texas Medical Branch, Galveston, TX 77555-1019, USA

**Keywords:** coronaviruses, nsp1, host gene expression suppression, mRNA degradation, translational suppression, virulence factor

## Abstract

Many viruses disrupt host gene expression by degrading host mRNAs and/or manipulating translation activities to create a cellular environment favorable for viral replication. Often, virus-induced suppression of host gene expression, including those involved in antiviral responses, contributes to viral pathogenicity. Accordingly, clarifying the mechanisms of virus-induced disruption of host gene expression is important for understanding virus–host cell interactions and virus pathogenesis. Three highly pathogenic human coronaviruses (CoVs), including severe acute respiratory syndrome (SARS)-CoV, Middle East respiratory syndrome (MERS)-CoV, and SARS-CoV-2, have emerged in the past two decades. All of them encode nonstructural protein 1 (nsp1) in their genomes. Nsp1 of SARS-CoV and MERS-CoV exhibit common biological functions for inducing endonucleolytic cleavage of host mRNAs and inhibition of host translation, while viral mRNAs evade the nsp1-induced mRNA cleavage. SARS-CoV nsp1 is a major pathogenic determinant for this virus, supporting the notion that a viral protein that suppresses host gene expression can be a virulence factor, and further suggesting the possibility that SARS-CoV-2 nsp1, which has high amino acid identity with SARS-CoV nsp1, may serve as a major virulence factor. This review summarizes the gene expression suppression functions of nsp1 of CoVs, with a primary focus on SARS-CoV nsp1 and MERS-CoV nsp1.

## 1. Introduction

Three highly pathogenic human coronaviruses (HCoVs), including severe acute respiratory syndrome (SARS)-CoV, Middle East respiratory syndrome (MERS)-CoV, and SARS-CoV-2, emerged within the first 20 years of the twenty-first century. The global epidemic of SARS-CoV in 2002–2003 revealed for the first time that HCoVs can present a significant global public health threat [1,2,3]. The mortality rate of SARS-CoV was ~10%, and the epidemic was contained within 2003 primarily due to strict isolation of affected individuals. About ten years after the SARS-CoV epidemic, MERS-CoV was identified from a man in Saudi Arabia, who died of acute pneumonia and renal failure [4]. MERS-CoV has higher mortality rates (~35%) than SARS-CoV and still remains a public health problem, mainly in the Arabian Peninsula [5]. Infections from MERS-CoV also occurred in countries outside of the Arabian Peninsula due to travel by infected persons [6]. The global pandemic of SARS-CoV-2, which was initially identified late in 2019 in Wuhan, China [7], is currently in progress. There has been substantial effort worldwide devoted to controlling the SARS-CoV-2 pandemic and the treatment of patients with COVID-19, the disease it causes.

CoVs are members of the subfamily *Coronavirina* within the family *Coronaviridae* and the order *Nidovirales*. They are classified into four genera, including Alphacoronavirus (α-CoV), Betacoronavirus (β-CoV), Gammacoronavirus (γ-CoV), and Deltacoronavirus (δ-CoV) [8,9,10,11]. α- and β-CoVs infect only mammals and γ- and δ-CoVs primarily infect birds, although some γ- and δ-CoVs can also infect mammals [8]. It has been well-recognized that CoVs are major pathogens for livestock, causing substantial economic losses [12,13,14]. They also infect mammalian pets, laboratory animals and many other wild animals [15]. All highly pathogenic HCoVs, including SARS-CoV, MERS-CoV, and SARS-CoV-2, belong to the genus β-CoV and are considered to be initially derived from wild mammals, most probably bats [16,17]. They are associated with severe lower respiratory tract infections [18,19,20], while other HCoVs, including HCoV-OC43 and HCoV-HKU1, belonging to the genus β-CoV, and HCoV-229E and HCoV-NL63, belonging to the genus α-CoV, cause relatively mild upper respiratory tract infections [21,22].

CoVs are enveloped RNA viruses that contain a large (~30 kb) capped and polyadenylated positive-sense and single-stranded RNA genome. The CoV particle comprises at least four canonical structural proteins, including N, S, M, and E proteins. The 5′ two-thirds of the genome encodes gene 1 proteins and the one-third at the 3′ end encodes structure and accessory proteins. The latter proteins are not required for virus replication in cell cultures [23,24]. Upon entry into host cells, the CoV genomic RNA, which is released into the cytoplasm, translates two large, partially overlapping precursor polyproteins from gene 1. Two virally encoded proteinases proteolytically process these precursor polyproteins to generate 16 mature proteins, labeled nonstructural protein 1 (nsp1) to nsp16 for α- and β-CoVs [25]. All of these gene 1 proteins, except for nsp1 [26] and nsp2 [27], are considered to be essential for viral RNA synthesis [28]. The intracellular form of the genome, mRNA 1, and several subgenomic mRNAs [29,30], all of which form a 3′ co-terminal nested structure and carry the same leader sequence of 60–70 nucleotides at the 5′ end, accumulate in infected cells [31,32,33]. Viral structural proteins and accessory proteins are translated from these subgenomic mRNAs. After accumulation of viral structural proteins and mRNA 1, assembly and budding of virus particles occurs at endoplasmic reticulum Golgi intermediate compartment membranes [34,35,36]. The virus particles are then released from the cells, and a recent study showed that β-CoV uses lysosomal trafficking for virus egress [37]. CoV M protein plays a critical role in virus assembly [38,39,40,41,42,43,44]. In many CoVs, E protein, which exists in low abundance in the virion, is necessary for production of high titers of infectious CoV particles [38,45,46,47].

Nsp1, one of the proteins encoded by gene 1, is encoded in only α- and β-CoVs [9,10,48,49,50]. The amino acid lengths of CoV nsp1 vary among genera and groups (Table 1). A phylogenetic tree of nsp1 shows some amino acid sequence diversity depending on genera and groups (Figure 1). Structural analyses showed high similarity in the core structure of the nsp1 of α- and β-CoVs, although amino acid sequence homology between the nsp1 of α-CoV and β-CoV is low [51]. The nsp1s of α- and β-CoVs share a biological function to inhibit host gene expression (Table 1) [52,53,54,55,56], but they use different strategies to exert this function [56,57,58,59,60].

**Table 1 cells-10-00300-t001:** Summary of the functions of coronavirus (CoV) nonstructural protein 1 (nsp1).

Nsp1		Group	Length (Amino Acid)	Functions/Characteristics	Reference
**β-CoV**	Mouse hepatitis virus (MHV)	2a	245	• Induction of cell cycle arrest	[61]
				• Inhibition of reporter gene expression	[53,62] *
				• Inhibition of type I interferon (IFN) signaling and production	[53,62] *
				• Pathogenic determinant in mice	[53,62] *
Severe actute respiratory syndrome		2b	180	• Inhibition of type I IFN induction and signaling	[55,63,64] *
(SARS)-CoV				• Translation inhibition	[57,59,63] *
				• Induction of host mRNA cleavage	[58,59]
				• Induction of chemokines and cytokines	[65,66]
				• Pathogenic determinant in mice	[67] *
				• Induction of cell cycle arrest	[64]
				• Disruption of localization of Nup93 from nuclear pore complex	[68]
SARS-CoV-2		2b	180	• Translation inhibition	[69,70,71,72]
				• Inhibition of type I IFN induction, signaling, and production	[69,71,73,74] *
				• Induction of cell cycle arrest	[75]
Middle East respiratory syndrome		2c	193	• Translation inhibition	[56]
(MERS)-CoV				• Induction of host mRNA decay	[56]
				• Promotion of virus assembly/budding in specific cell line	[76] *
Bat CoV: Rm1		2b	180	• Inhibition of host protein synthesis	[54]
Bat CoV:133		2c	193	• Induction of host mRNA decay	
Bat CoV:HKU9-1		2d	175	• Inhibition of type I IFN and IFN-stimulated gene induction	
**α-CoV**	Transmissible gastroenteritis virus	1a	110	• Suppression of host protein synthesis/Translation inhibition	[52,60]
(TGEV)				• Pathogenic determinant in pigs	[52] *
				• Inhibition of type I IFN induction and signaling	[77]
				• Induction of cell cycle arrest	[77]
Feline infectious peritonitis virus (FIPV)		1a	110	• Inhibition of reporter gene expression	[52]
				• Inhibition of IFN-stimulated gene expression	[77] *
				• Inhibition of type I IFN induction and signaling	[77]
				• Induction of cell cycle arrest	[77]
Porcine epidemic diarrhea virus (PEDV)		1b	110	• Suppression of host protein synthesis	[51]
				• Suppression of type I IFN induction and signaling	[77,78,79]
				• Suppression of type III IFN induction	[80]
				• Induction of cell cycle arrest	[77]
Swine acute diarrhea syndrome		1b	110	• Inhibition of type I IFN induction and signaling	[77]
(SADS)-CoV				• Induction of cell cycle arrest	[77]
Human CoV (HCoV)-229E		1b	110	• Inhibition of reporter gene expression	[52,53,81]
				• Inhibition of type I IFN induction, signaling, and production	[77,81] *
				• Induction of cell cycle arrest	[77]
HCoV-NL63		1b	110	• Inhibition of reporter gene expression	[52,81]
				• Inhibition of type I IFN induction, signaling, and production	[77,81] *

* denotes function(s) validated in virus infection (e.g., Sendai virus infection; Newcastle disease virus infection; recombinant CoV infection).

In this review, we outline key studies that have delineated the mechanisms underlying host gene expression inhibition by CoV nsp1, mainly SARS-CoV nsp1 and MERS-CoV nsp1, and briefly SARS-CoV-2 nsp1.

## 2. Experimental Approaches that Are Used to Explore Biological Functions of CoV Nsp1

Several different experimental approaches have been used to study biological functions of CoV nsp1. One initial approach was expressing nsp1 in cultured cells [52,53,54,55,56,60,63,71], which led to the discovery of the nsp1-induced suppression of host gene expression. Reverse genetics systems of CoVs serve as valuable tools to explore the biological significance of CoV nsp1 for viral replication and pathogenicity [52,53,62,63,64,67,76]. Another experimental approach for understanding the biological functions of nsp1 is by using cell-free in vitro assay systems, where purified CoV nsp1, which is first expressed from *E. coli* or insect cells, is used to suppress gene expression of in vitro-synthesized RNA transcripts in rabbit reticulocyte lysate (RRL) or HeLa cell extracts [57,58,59,60]. The cell-free assay systems allow easy manipulation of the experimental conditions, e.g., adding specific reagents or removing specific cellular factors, which contributes to a better understanding of the molecular mechanisms of gene expression suppression by CoV nsp1. An example of successful use of a cell-free in vitro assay system was the discovery of an essential role of ribosomes for induction of SARS-CoV nsp1-induced endonucleolytic cleavage of nonviral mRNAs (see below) [59]. To date, cell-free in vitro assay systems have been established for SARS-CoV nsp1, transmissible gastroenteritis virus (TGEV) nsp1, and SARS-CoV-2 nsp1 [59,60,69].

## 3. CoV Nsp1 Serves as a Pathogenic Determinant and an Inhibitor of Antiviral Gene Expression

Nsp1 is a major virulence factor in several CoVs [52,53,67]. Zest et al. showed that expression of nsp1 of mouse hepatitis virus (MHV), a β-CoV, reduces cellular gene expression [53]. By using wild-type MHV and a mutant MHV encoding a deletion in the nsp1-coding sequence, the authors showed that MHV nsp1 acted as an interferon (IFN)-antagonist and was a major virulence factor for mice. Subsequently, Lei et al. reported that a specific amino acid region of MHV nsp1, LLRKxGxKG at position 191 to 199, is important for the pathogenicity in mice [62].

Like MHV nsp1, SARS-CoV nsp1 plays an important role in suppressing antiviral responses [55,63,64]. Also, nsp1 of mouse-adapted SARS-CoV is a pathogenic determinant in mice [67]. The amino acid region of SARS-CoV nsp1, which corresponds to the LLRKxGxKG region of MHV nsp1, is responsible for pathogenicity in mice and inhibition of host antiviral signaling pathways [64], suggesting the importance of the region corresponding to the LLRKxGxKG region of MHV nsp1 for virulence of β-CoVs (Figure 2). Distinct but overlapping regions of nsp1, which interact with different host factors, may exert the inhibition of host gene expression and antiviral signaling pathways [83]. Expression of SARS-CoV nsp1 and SARS-CoV replication enhances signaling through the Calcineurin/NFAT (nuclear factor of activated T cells) pathway, which is important for immune cell activation [65], and SARS-CoV nsp1 expression induced the secretion of several chemokines in human lung epithelial cells [66], implying a possible role for nsp1 in immune dysregulation.

SARS-CoV nsp1 also disrupts the nuclear-cytoplasmic transport of biomolecules [68]. SARS-CoV nsp1 associates with Nup93, a member of the nuclear pore complex, and displaces it from the nuclear pore complex [68]. SARS-CoV nsp1 expression alters the nuclear-cytoplasmic distribution of nucleolin, which is an RNA-binding protein found primarily in the nucleus. These studies imply that SARS-CoV nsp1 affects multiple steps in expression of host genes, including antiviral genes.

Like nsp1 of β-CoVs, nsp1 of α-CoVs serves as an inhibitor of antiviral gene expression. HCoV-229E nsp1 reduces luciferase reporter gene expression under the control of the IFN-β- and IFN-stimulated response element [53]. Nsp1 of porcine epidemic diarrhea virus (PEDV) uses multiple mechanisms to suppress host innate immune responses: it interrupts the enhanceosome assembly of IRF3 and CREB-binding protein by degrading the latter protein, resulting in suppression of type I IFN production [78], it is a potent NF-κB antagonist by inhibiting phosphorylation and subsequent degradation of IκBα, leading to suppression of IFN production and early production of pro-inflammatory cytokines [79], and it blocks the nuclear translocation of IRF1 and reduces the number of peroxisomes to suppress IRF1-mediated induction of type III IFNs [80]. Recently, Shen et al. reported that the conserved region (amino acids 91–95) of nsp1 of α-CoVs, including feline infectious peritonitis virus (FIPV), TGEV, PEDV, HCoV-229E, and HCoV-NL63, is responsible for suppression of host gene expression [52]. Notably, they also demonstrated that TGEV nsp1 plays a critical role in viral virulence in pigs. Shen et al. also demonstrated that expression of nsp1 of TGEV, porcine respiratory coronavirus (PRCV), swine acute diarrhea syndrome coronavirus (SADS-CoV), PEDV, HCoV-229E, or HCoV-NL63 reduces IFN-related gene expression, expression of the nsp1s of the α-CoVs markedly downregulates STAT1 phosphorylation at S727 residue without affecting the STAT1 expression levels and STAT1 phosphorylation at S701, and *STAT*1, *ISG*15, and *IRF*9 mRNAs are significantly upregulated in cells infected with a FIPV mutant carrying a deletion of amino acids 91–95 in the nsp1 without severely affecting virus replication ability [77]. These data suggest that nsp1 proteins of α-CoVs also act as major pathogenic determinants like that of β-CoVs.

## 4. Biological Functions of SARS-CoV Nsp1 and SARS-CoV-2 Nsp1

### 4.1. Translational Suppression Induced by SARS-CoV Nsp1 and SARS-CoV-2 Nsp1

The finding that SARS-CoV nsp1 expression suppresses reporter gene expression from co-transfected plasmids in mammalian cells represents the first demonstration of the gene expression suppression function of CoV nsp1 [55]. By using bacterially expressed SARS-CoV nsp1 in an in vitro translation system, Kamitani et al. demonstrated that SARS-CoV nsp1 stably binds to the 40S ribosomal subunit and inactivates translational function of the 40S ribosomal subunit, leading to inhibition of protein synthesis [59]. Consistent with this finding, SARS-CoV nsp1 displays cytoplasmic localization in infected cells and in expressed cells [55]. The K164A and H165A substitutions near the C-terminal region of SARS-CoV nsp1 abolish its 40S subunit-binding function and translation inhibition activity, demonstrating the importance of K164 and H165 residues for the translational suppression function of the SARS-CoV nsp1 (Figure 2) [59]. A subsequent study revealed that SARS-CoV nsp1 inhibits the translation initiation step by targeting at least two separate stages: one is 48S initiation complex formation and the other is the step(s) that is involved in the 80S initiation complex formation from the 48S complex [57].

SARS-CoV-2 nsp1 has 84% amino acid sequence identity with SARS-CoV nsp1 (Figure 2), suggesting that both proteins have similar biological functions. Thoms et al. revealed that SARS-CoV-2 nsp1 binds to 40S and 80S ribosome subunits and disrupts cap-dependent translation [69]. Like SARS-CoV nsp1, the K164 and H165 residues close to the C-terminus of SARS-CoV-2 nsp1 are important for ribosome binding and translation inhibition (Figure 2) [69]. To understand the molecular interactions of SARS-CoV-2 nsp1 with human ribosomes, the authors reconstituted a complex made by human 40S ribosomal subunits and nsp1 and examined the structure of the complex by cryo-electron microscopy. Their study demonstrated that the C-terminal region of SARS-CoV-2 nsp1 interacts with the rRNA helix h18 and two ribosomal proteins, uS35 and uS3, to establish tight nsp1–40S subunit interaction. Surface of the C-terminal region of the nsp1 carries three major patches, each of which interacts with rRNA or ribosomal proteins via matching surface charges. Furthermore, the shape of the C-terminal region of nsp1 matches the shape of the mRNA channel and completely obstructs accommodation of regular mRNA to the mRNA channel, leading to strong nsp1-induced inhibition of translational initiation. By using similar experimental methods, studies by Schubert et al. led to the same conclusions [70]. Banerjee et al. also reported that nsp1 binds to 18S rRNA in the mRNA entry channel of the 40S ribosome and leads to translational inhibition [71].

### 4.2. Effects of SARS-CoV Nsp1 on Stabilities of Host RNAs and Expression of Innate Immune Genes

In addition to inhibiting translation by binding to 40S ribosomes, SARS-CoV nsp1 also induces degradation of endogenous host mRNAs in expressed cells and in SARS-CoV-infected cells [55,63], further contributing to host gene expression suppression. The complex of SARS-CoV nsp1-40S ribosome induces the endonucleolytic cleavage at the 5′ region of capped nonviral mRNAs and renders the mRNA translationally incompetent [58,59]. In the absence of 40S subunits, SARS-CoV nsp1 cannot induce the endonucleolytic cleavage of capped mRNA templates [59], suggesting that the nsp1 that binds to the 40S submit, but not free nsp1, can induce the RNA cleavage. Only Pol II transcripts, i.e., translationally active mRNAs, but not Pol III or Pol I transcripts, are degraded by SARS-CoV nsp1 [84], further supporting the importance of nsp1-binding to 40S ribosomes for the induction of mRNA cleavage. Host exonuclease Xrn I degrades the mRNAs that are cleaved by the SARS-CoV nsp1 [84]. SARS-CoV nsp1 also induces RNA cleavage within the ribosome loading region of type I and type II picornavirus internal ribosome entry site (IRES) elements, whereas it does not induce RNA cleavage within the IRES elements of hepatitis C virus or cricket paralysis virus, demonstrating that the nsp1-induced RNA modification is template-dependent [59]. Many SARS-CoV nsp1-induced cleavage sites in capped mRNA transcripts are detected within 30 nt of the 5′ untranslated region [58]. The ribosome footprint on a mRNA is ~28 nt [85,86], demonstrating that the cleavage sites of the template mRNA transcripts are either in or proximal to the initial ribosome binding sites. These data imply that SARS-CoV nsp1-induced endonucleolytic RNA cleavage occurs as soon as the 43S preinitiation complex containing a nsp1-bound 40S subunit binds to the 5′ cap-proximal region of an mRNA.

Because binding of nsp1 to 40S ribosomal subunit is required for the nsp1-induced endonucleolytic mRNA cleavage [59] and SARS-CoV nsp1 has no similarities in its primary amino acid sequence or protein structure with any known host proteins, including RNases [87], it has been hypothesized that SARS-CoV nsp1 uses a host endonuclease to induce endonucleolytic cleavage of template mRNA transcripts that interact with 40S ribosomes. The identity of this putative host RNase, which is recruited and/or activated by SARS-CoV nsp1, is unknown. Meanwhile, R125A and K126A substitutions at the surface of SARS-CoV nsp1 abolish the endonucleolytic RNA cleavage activity [57], implying the importance of these residues for recruitment and/or activation of the putative host endonuclease (Figure 2).

By using a reverse genetics system, Narayanan et al. generated wild-type SARS-CoV and a SARS-CoV mutant virus (SARS-CoV-mt) carrying nsp1 with the K164A and H165A mutations, which abolishes the nsp1′s host gene suppression functions. Both viruses exhibited similar one-step growth kinetics and accumulated similar levels of viral mRNAs and nsp1 protein in infected cells, whereas SARS-CoV-mt-infected cells showed clearly higher amounts of endogenous host mRNAs than SARS-CoV-infected cells, in the presence or absence of actinomycin D [63]. Furthermore, host protein synthesis inhibition in SARS-CoV-infected cells was stronger than in SARS-CoV-mt-infected cells. These data demonstrated that SARS-CoV nsp1 promotes degradation of host mRNAs and suppresses host protein synthesis in SARS-CoV-infected cells. These data also implied that SARS-CoV nsp1 may not induce viral mRNA degradation. They also revealed that SARS-CoV-mt replication, but not SARS-CoV replication, induces efficient accumulations of mRNAs of IFN-β, ISG15, and ISG56, the latter two of which are IFN-inducible genes, as well as efficient production of type I IFN [63]. These data demonstrated that SARS-CoV nsp1 suppresses host innate immune responses, including type I IFN expression, in SARS-CoV-infected cells. The data that show the amounts of *IFN-*β, *ISG*15, and *ISG*56 mRNAs are higher in cells infected with SARS-CoV-mt than in those infected with SARS-CoV suggest that the RNA cleavage function of nsp1 contributes to suppression of host innate immune functions. Meanwhile, how efficiently the nsp1-mediated translational suppression function inhibits host innate immune functions has not been experimentally examined. Generation and characterization of a SARS-CoV mutant carrying nsp1 R125A and K126A substitutions, which lacks the RNA cleavage function but retains the translation inhibition function, would reveal the importance of nsp1′s translation inhibition function for suppression of host innate immune functions.

Recent studies demonstrated that like SARS-CoV nsp1 [55,63,64], expressed SARS-CoV-2 nsp1 suppresses production of IFN-β [69,73,74].

### 4.3. Effects of Nsp1 from SARS-CoV and SARs-CoV-2 on Viral Gene Expression

Virological characterization of wild-type SARS-CoV and SARS-CoV-mt initially suggested that SARS-CoV nsp1 may not induce viral mRNA degradation [63]. Further studies showed that unlike many non-viral mRNAs, SARS-CoV nsp1 does not induce endonucleolytic cleavage in capped and polyadenylated SARS-CoV-like mRNA that contains the SARS-CoV leader sequence in an in vitro assay [58]. Importantly, SARS-CoV nsp1 can induce RNA cleavage in SARS-CoV-like mRNA that lacks the leader sequence, demonstrating that it is the leader sequence that protects the viral mRNAs from the nsp1-induced endonucleolytic RNA cleavage. SARS-CoV-like mRNA carrying an extra G residue at the 5′ end and another virus-like mRNA, carrying a U to G substitution at the second nucleotide position, were susceptible to the nsp1-induced endonucleolytic RNA cleavage [58], implying that the very 5′-terminal region of the SARS-CoV leader sequence is important for evasion from the nsp1-induced RNA cleavage. Tanaka et al. reported that SARS-CoV nsp1 bound to stem loop (SL) 1, one of four helical SL structures in the SARS-CoV leader sequence, and protected the SARS-CoV mRNAs from nsp1-mediated RNA degradation [88]. Amino acid R124 of SARS-CoV nsp1 is responsible for recognizing SARS-CoV mRNA and protecting it from nsp1-induced RNA cleavage (Figure 2). Their SARS-CoV replicon system also showed the importance of the specific interaction of nsp1 with the viral mRNAs for efficient SARS-CoV replication.

Although SARS-CoV nsp1 binds to 40S ribosomes and suppresses translation of any mRNAs, the data that show that SARS-CoV and SARS-CoV-mt showed similar one-step growth kinetics [63] led Narayanan et al. to examine the possibility that SARS-CoV mRNAs may escape from nsp1-induced translation inhibition by spatially separating from nsp1 in infected cells. RNA fluorescence in situ hybridization and confocal microscopic analyses of viral mRNAs and nsp1 in SARS-CoV-infected cells showed that viral mRNAs are not spatially separated from nsp1 [89]. Although SARS-CoV and SARS-CoV-mt showed similar virus replication kinetics and viral mRNA accumulation, viral structural and accessory proteins were accumulated at higher levels in SARS-CoV-infected cells than in SARS-CoV-mt-infected cells [89]. These data revealed that nsp1 inhibits viral translation in SARS-CoV-infected cells and further indicated that the reduced levels of viral proteins in infected cells are still sufficient for optimal virus replication.

For SARS-CoV-2, it has been reported that the presence of a viral leader sequence at the 5′ end of expressed non-viral transcripts facilitates escape of the transcripts from nsp1-mediated suppression of gene expression [70,71]. The cis-acting RNA hairpin SL1 in the leader sequence of SARS-CoV-2 facilitates evasion of viral RNAs from the nsp1-induced translational suppression [71,72]. Banerjee et al. proposed a model where the viral leader sequence allosterically modulates nsp1 structure, leading to potential dissociation of the nsp1 from the 40S subunits and allowing viral translation [71], while Tidu et al. showed that nsp1 remains bound on the ribosome during viral translation and proposed that the interaction between nsp1 and SL1 frees the mRNA accommodation channel while maintaining nsp1 bound to the ribosome [72]. It should be noted that data shown by Tidu et al. revealed that nsp1 suppressed translation of viral mRNAs in RRL, the extent of which was less efficient than IRES-mediated translation [72]; hence, like SARS-CoV nsp1 [57], SARS-CoV-2 nsp1 suppresses gene expression of viral mRNAs in RRL. If SARS-CoV-2 nsp1 induces endonucleolytic cleavage of nonviral mRNAs, but not viral mRNAs, experiments using a SARS-CoV-2 nsp1 mutant that suppresses translation, but lacks RNA cleavage activity, would help to properly interpret data testing the effects of nsp1 on translation of host and viral mRNAs. Accordingly, it would be important to test whether SARS-CoV-2 nsp1 induces endonucleolytic cleavage of various host and viral mRNAs.

## 5. Biological Functions of MERS-CoV Nsp1

### 5.1. Effects of MERS-CoV Nsp1 on Host and Viral Gene Expressions

Although amino acid identity between SARS-CoV nsp1 and MERS-CoV nsp1 is ~22%, both nsp1 proteins share the property of inhibiting host gene expression at posttranscriptional levels. Namely, MERS-CoV nsp1 inhibits host translation and induces mRNA degradation in cells expressing nsp1 and in MERS-CoV-infected cells [56,76]. SARS-CoV nsp1 carrying R125A and K126A substitutions is unable to induce endonucleolytic RNA cleavage in host mRNAs, however, the mutant still retains the translation suppression function [57]. Likewise, a MERS-CoV nsp1 mutant carrying R146A and K147A substitutions, which correspond to the R125A and K126A substitutions in SARS-CoV nsp1, is competent for host translational suppression and is deficient for inducing mRNA cleavage and degradation (Figure 2) [56]. These data indicate that the mRNA degradation activity of MERS-CoV nsp1, which is most probably triggered by nsp1-induced endonucleolytic RNA cleavage, followed by Xrn I-mediated digestion of the cleaved RNA from the 5′ to the 3′ direction, is separable from its translation inhibitory function.

Despite these functional similarities between SARS-CoV nsp1 and MERS-CoV nsp1, they suppress host translation by using different strategies. SARS-CoV nsp1 is localized in the cytoplasm, as it binds to the 40S ribosomal subunits to gain access to translating mRNAs [55,59]. In contrast, MERS-CoV nsp1 does not stably bind to the 40S ribosomal subunits and is detected in the nucleus and the cytoplasm [56]. Interestingly, MERS-CoV nsp1 selectively targets mRNAs, which are synthesized in the nucleus and transported to the cytoplasm, to induce translation inhibition and mRNA degradation, but spares virus-like mRNA that originates in the cytoplasm or exogenous mRNA transcripts that are directly introduced into the cytoplasm [56]. These findings show a unique viral strategy, in which the cytoplasmically synthesized MERS-CoV mRNAs can escape from the inhibitory effects of MERS-CoV nsp1. Although how MERS-CoV nsp1 selectively targets nucleus-derived host mRNAs is unknown, MERS-CoV nsp1 may selectively target nucleus-derived mRNPs through its interaction with one of the cellular mRNA-binding proteins that form the host mRNP complex and inhibit the expression of host genes (Figure 3).

Similar to SARS-CoV nsp1, Terada et al. reported that interaction between MERS-CoV nsp1 and SL1 in the leader sequence of MERS-CoV mRNAs is important for the escape of the viral mRNAs from MERS-CoV nsp1-mediated shutoff and efficient virus replication [90]. They also showed that amino acid residue R13 in MERS-CoV nsp1 is responsible for recognizing MERS-CoV mRNAs and protecting them from the nsp1-mediated shutoff (Figure 2). Exactly how this nsp1–SL1 interaction promotes virus replication requires further investigation, yet these data imply that the specific interaction between MERS-CoV nsp1 and the SL1 in the leader sequence could serve as a potential therapeutic target against MERS-CoV.

### 5.2. Biological Significances of MERS-CoV Nsp1 for Viral Replication

Nakagawa et al. examined the biological functions of MERS-CoV nsp1 in virus replication using a parental wild-type MERS-CoV and two mutant viruses with specific mutations in nsp1: one carrying the K181A mutation, which abolishes the RNA cleavage function and the translation inhibition function (MERS-CoV-mt) (Figure 2), while the other carried K164A and K165A mutations, which ablate the RNA cleavage function but not the translation inhibition function (MERS-CoV-CD) [76]. All three viruses efficiently replicated with similar replication kinetics in Vero cells, while the mutant viruses showed more attenuated host translational suppression and host mRNA degradation activities than the parental virus. These data demonstrated that MERS-CoV nsp1 suppresses host gene expression in infected cells.

They also reported that the parental virus replicated more efficiently than the mutant viruses in Huh-7 cells, 293-derived cells (293/CD26 cells), and HeLa-derived cells, the latter two of which stably expressed a viral receptor protein, dipeptidyl peptidase 4 [91], also called CD26, suggesting that MERS-CoV nsp1 promotes virus replication in a cell type-dependent manner. Replication of the three viruses in 293/CD26 cells resulted in accumulation of similar levels of nsp1 and major viral structural proteins, without inducing *IFN*-β and *IFN*-λ mRNAs, implying that inefficient replication of the mutant viruses was not due to induction of type I and III IFNs. Unlike SARS-CoV nsp1, the host gene suppression functions of MERS-CoV nsp1 may not have a critical role in the suppression of host innate immune responses in MERS-CoV-infected cells. Interestingly, the parental MERS-CoV replicated to a higher titer than the two mutant viruses in 293/CD26 cells, whereas the three viruses accumulated comparable amounts of viral structural proteins in infected cells. The cell-associated virus titers and electron microscopic analysis of infected cells revealed that both mutant viruses do not generate intracellular virus particles as efficiently as the parental virus, leading to inefficient infectious virus production. Nsp1 is not detectable in purified MERS-CoV and the mutant virus particles, suggesting that MERS-CoV nsp1 does not promote virus assembly and/or budding by incorporating itself into virions. These data strongly indicate that the RNA cleavage function of MERS-CoV nsp1 promotes virus assembly and/or budding in 293/CD26 cells. MERS-CoV nsp1 represents the first CoV gene 1 protein that has an important role in efficient virus assembly and/or budding and is the first identified viral protein whose RNA cleavage-inducing function promotes virus assembly and/or budding.

## 6. CoV Nsp1-Induced Cell Cycle Arrest

Like many DNA viruses, which usurp host cell cycle regulation for their own replication advantage [92], some RNA viruses, such as Coxsackievirus A6 and murine norovirus-1, induce cell cycle arrest in G0/G1 phase to provide cellular environment that is suitable for efficient viral production [93,94]. It is known that nonstructural 3D protein of Enterovirus D68 arrests the cell cycle at the G0/G1 phase to promote viral production [95]. In the case of CoVs, Chen et al. demonstrated that expressed MHV nsp1 induces G0/G1 cell cycle arrest [61]. Their data suggest that MHV nsp1 expression stabilizes tumor suppressor p53, leading to transcription of cycline-dependent kinase inhibitor p21^CipI^. The increased amounts of p21^CipI^ inhibits cyclin E/Cdk2 activity, resulting in inhibition of hyperphosphorylation of retinoblastoma protein and prevention of cell cycle progression from the G0/G1 to S phase. Expression of SARS-CoV nsp1also inhibits cell proliferation and increases the number of cells in G0/G1 phase without affecting cell viability [64]. Likewise, SARS-CoV-2 nsp1 expression induces G0/G1 phase arrest [75]. Also, nsp1 of α-CoV, including TGEV, PRCV, SADS-CoV, PEDV, HCoV-229E, or HCoV-NL63, arrests the cell cycle at the G0/G1 phase in expressing cells [77]. These data indicate a potential common role of CoV nsp1 for arresting cells in the G0/G1 phase, which may be important for optimal CoV production.

## 7. Mechanism of Host Gene Expression Inhibition by Nsp1 of α-CoVs

Although our understandings of the biological functions of α-CoV nsp1 are somewhat limited, several studies have revealed the biological functions of TGEV nsp1. Expression of TGEV nsp1 strongly inhibits reporter gene expression and host protein synthesis in mammalian cells [60]. Expressed TGEV nsp1 is detected in both the nucleus and the cytoplasm [89]. TGEV nsp1 neither binds the 40S ribosomal subunit nor promotes host mRNA degradation, indicating that TGEV nsp1 suppresses host gene expressing by using a mechanism that differs from SARS-CoV nsp1. Interestingly, TGEV nsp1 inhibits protein translation in HeLa cell extracts, whereas it does not affect translation in RRL. Furthermore, TGEV nsp1 suppresses translation in RRL supplemented with HeLa S100 post-ribosomal supernatant or HeLa S10 extract [60]. These data suggest that RRL lacks a factor(s) that is needed for TGEV nsp1-mediated translational inhibition and that HeLa S10 extract and post-ribosomal HeLa S100 supernatant contain this putative factor. Inactivation of TGEV nsp1-mediated translation inhibition activity does not affect virus replication, yet it significantly reduces virus virulence in piglets [52], strongly supporting the possibility that nsp1 of α-CoV is a major pathogenic determinant.

Expression of HCoV-229E nsp1 or HCoV-NL63 nsp1 in mammalian cells inhibits reporter gene expression driven by SV40, HSV-TK, or CMV promoters [53,81]. Shen et al. reported that a conserved region (amino acid position at 91–95) of nsp1 from α-CoVs, including FIPV, PEDV, HCoV-229E, HCoV-NL63, and TGEV, is important for their function of host gene expression inhibition [52].

## 8. Perspective

As CoV nsp1 is a major viral virulence factor, surveying emergence and spread of HCoV variants carrying nsp1 mutations in the current SARS-CoV-2 pandemic and possible future pandemics caused by novel HCoVs becomes important. These efforts would help reveal the natural evolution and adaptation of HCoVs in human hosts and provide clues as to potential changes in pathogenesis of variants emerging during pandemics. Benedetti et al. identified a SARS-CoV-2 genome carrying a deletion of 9 nucleotides in position 686–694, corresponding to the amino acid residues of K141, S142, and F143 close to the C-terminus of nsp1 from patients of several countries [96]. As the C-terminal region of SARS-CoV-2 interacts with 40S ribosome subunits [69,70,71], the newly identified deletion in the KSF amino acids may potentially influence nsp1 structure, abolishing the binding the nsp1 mutant to 40S ribosomes, and affecting activity on viral and host gene expression regulation. Consistent with this supposition, substitution of two of these amino acids (KS) reverts the loss of IFN-α expression in cells expressing mutated SARS-CoV nsp1 [83]. These data suggest that the nsp1 gene of the SARS-CoV-2 genome is undergoing an evolutionary process, which may result in better adaptation of the virus to human hosts.

There has been rapid and significant progress in understanding the high-resolution structure of the complex of 40S subunits and SARS-CoV-2 nsp1 [69,70]. These structural based-insights into nsp1-40S ribosome interactions may facilitate rational design of antiviral drugs targeting the SARS-CoV-2–nsp1–ribosome interaction. One caveat, however, is that antiviral drugs targeting the C-terminal region of nsp1, in an effort to block nsp1–40S ribosome interactions, could generate SARS-CoV-2 escape mutants carrying mutated nsp1, to which the antiviral drugs no longer bind. The mutant viruses would replicate in infected hosts, as nsp1 is considered to be not essential for virus replication. Emergence of naturally occurring SARS-CoV-2 variants carrying deletions of the KSF amino acids [96] further supports this possibility. It is worth noting that the 5′ end cis-acting RNA replication signal of MHV includes sequences encoding the N-terminal region of nsp1 gene [97]. If the 5′ end cis-acting RNA replication signal of SARS-CoV-2 also includes the N-terminal region of the nsp1 gene, development of antiviral drugs that abolish biological functions of nsp1 by binding to its N-terminal region would be meaningful, as escape mutants may not emerge easily. Namely, the escape mutant viruses carrying mutations in the nsp1-coding region within the 5′ end cis-acting RNA replication signal may not replicate efficiently due to alteration of the structure of the RNA replication signal, which is essential for efficient viral RNA synthesis.

The absence of nsp1 in γ- and δ-CoVs [9,48,49,50] raised a question as to how γ- and δ-CoVs undergo efficient replication without nsp1. Kint et al. showed that infectious bronchitis CoV (IBV), a γ-CoV, inhibits host translation, including type I IFN, without degradation of host mRNAs [98]. They further demonstrated that accessory protein 5b has a crucial role in IBV-induced host translation inhibition. Deletion of the IBV 5b gene results in attenuation in embryonated eggs without affecting fundamental viral replication ability [99], showing that IBV 5b is one of the pathogenic determinants, like nsp1, in α- and β-CoVs. These data suggest that IBV 5b serves biologically equivalent roles to nsp1 of α- and β-CoVs, yet it is unclear why IBV encodes a viral protein, whose function is similar to nsp1, as one of viral accessory proteins.

Several important questions remain to be addressed for further understanding of the biological functions of CoV nsp1. It has been speculated that nsp1 of SARS-CoV and MERS-CoV use a host RNase to induce endouncleolytic RNA cleavage to host mRNAs, yet this putative RNase has not been identified. It is also unclear how exactly the leader sequence protects SARS-CoV mRNAs from the RNA cleavage from this putative host RNase. MERS-CoV nsp1 displays an intriguing property of selectively targeting mRNAs of the nuclear origin for translation inhibition and mRNA degradation [56]. It has been speculated that MERS-CoV nsp1 accesses to nucleus-derived mRNAs, by binding to one of the mRNA-binding proteins that form the host mRNP complex, and inhibits gene expression (Figure 3), although the identity of this putative mRNA-binding protein remains unclear. Likewise, a factor(s), which is present in HeLa S10 extract and post-ribosomal HeLa S100 supernatant, but not in RRL, and plays a critical role in translational suppression of TGEV nsp1, has not been identified. Addressing these questions will certainly advance our understanding of the molecular mechanisms of nsp1-induced suppression of gene expression and the evasion mechanism of viral mRNAs from the nsp1 functions. The obtained data may potentially lead to a novel strategy for suppressing nsp1 function and inhibiting efficient replication of CoVs, which are currently causing public health and economic problems and continue to pose threats in the future.

## Figures and Tables

**Figure 1 cells-10-00300-f001:**
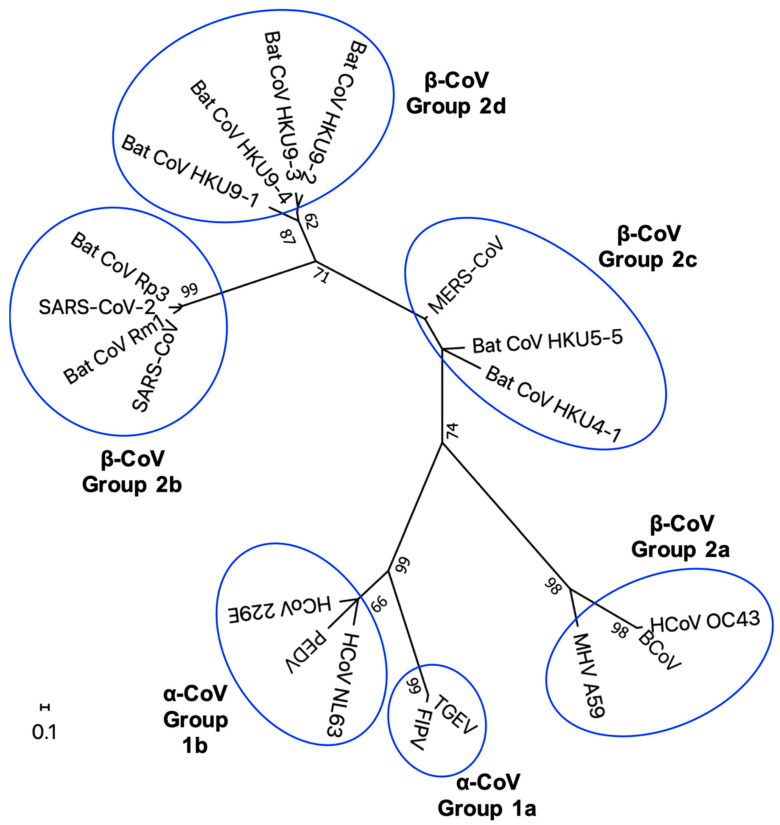
Phylogenetic tree of CoV nsp1. Unrooted maximum likelihood phylogeny of 17 publicly available amino acid sequences of CoV nsp1 is shown. a-CoVs and b-CoVs are divided into groups 1a and 1b and groups 2a, 2b, 2c, and 2d respectively, and the tree, based on 100 amino acid residues, was designed using MEGA X [82]. All positions containing gaps and missing data were eliminated. Bootstrap values (1000 replicates) above 60 are shown. The scale bar indicates the estimated number of substitutions per 10 amino acids. The virus strains used in the phylogenetic analysis are as follows: SARS-CoV-2 (accession no.: MN908947.3); SARS-CoV (Urbani strain, accession no.: AY278741.1); Bat CoV Rm1, bat isolates from *Rhinolophus macrotis* (horseshoe bat) (accession no.: DQ412043.1); Bat CoV Rp3, bat isolate from *R. pearsoni* (accession no.: DQ071615.1); Bat CoV HKU4-1, bat isolates from *Tylonycteris pachypus* (lesser bamboo bat) (accession no.: NC_009019.1); Bat CoV HKU5-5, bat isolate from *Pipistrellus abramus* (Japanese pipistrelle) (accession no.: EF065512.1); Bat CoV HKU9-1, Bat CoV HKU9-2, Bat CoV HKU9-3, and Bat CoV HKU9-4, bat isolates from *Rousettus leschenaulti* (Leschenault’s rousette) (accession no.: EF065513.1, EF065514.1, EF065515.1, EF065516.1 and, respectively); MHV strain A59 (accession no.: AF029248.1); human coronavirus OC43 (HCoV OC43) (accession no.: MN306036); bovine coronavirus (BCoV) (accession no.: NC_003045.1); feline infectious peritonitis virus (FIPV) (accession no.: MG893511.1); transmissible gastroenteritis virus (TGEV) (accession no.: KX900408.1); porcine epidemic diarrhea virus (PEDV) (accession no.: MT843278.1); human coronavirus NL63 (HCoV NL63) (accession no.: MN306018.1); human coronavirus 229E (HCoV 229E) (accession no.: KY369908.1).

**Figure 2 cells-10-00300-f002:**
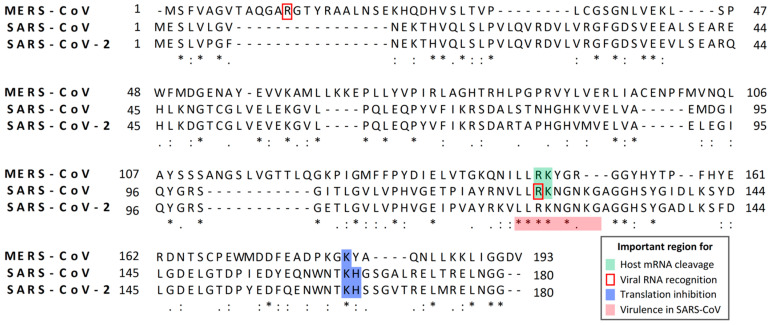
Alignment of the amino acid sequences of nsp1s of MERS-CoV, SARS-CoV, and SARS-CoV-2. Nsp1 sequences of the MERS-CoV strain EMC2012 (accession no.: YP_009047229), SARS-CoV strain Urbani (accession no.: AAP13442), and SARS-CoV-2 isolate Wuhan-Hu-1 (accession no.: MN908947.3) are aligned using Multiple Sequence Comparison by Log-Expectation (MUSCLE) alignment algorithm. Perfect matches, high-amino acid similarities, and low-amino acid similarities are represented by asterisks, double dots, and single dots, respectively. A dash “-” indicates a gap in the sequence. The numbers beside the aligned sequences show the positions of amino acid residues. Residues shown in green represent the functional amino acids for host mRNA cleavage. Residues shown by a red box outline represent the functional amino acids for viral mRNA recognition. Residues shown in blue represent the amino acids that are important for translation inhibition. Residues shown in red represent the amino acids that are important for virulence in mice.

**Figure 3 cells-10-00300-f003:**
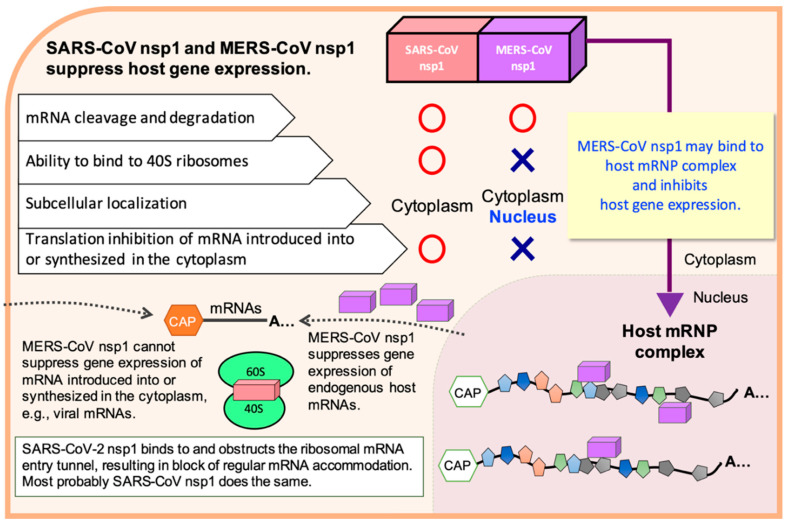
Biological properties of SARS-CoV nsp1, MERS-CoV nsp1, and SARS-CoV-2 nsp1 for inhibiting host gene expression and a possible mechanism of the MERS-CoV nsp1-mediated inhibition of host gene expression. It has been suspected that SARS-CoV nsp1 and MERS-CoV nsp1 recruit a cellular endonuclease to induce endonucleolytic RNA cleavage in host mRNAs, leading to accelerated mRNA degradation.

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
