# Peer review of "Mechanisms of Coronavirus Nsp1-Mediated Control of Host and Viral Gene Expression"

_cells, 2021, doi:10.3390/cells10020300_

Round 1

Reviewer 1 Report

Keisuke et al. provided a systematic summary of coronavirus nsp1. The authors introduced the functions and conservation of nsp1 in different genera of coronaviruses. nsp1 is a major virulence factor and plays an important role in suppressing antiviral responses. nsp1 can inhibit the host protein translation progress and induce the host mRNA endonucleolytic cleavage and degradation. nsp1 suppresses host innate immune functions, including type I IFN expression, to create a better environment for the virus replication. This review mainly focuses on SARS-CoV nsp1 and MERS-CoV nsp1, with brief descriptions of α-CoVs and γ-CoV. Overall this is a timely review with useful information to the general readers.

Giving the ongoing pandemic and hence the interest of the community, one particular area that can be improved is a more detailed description of the specifics of SARS-CoV-2 nsp1, beyond what has been learned from the original SARS-CoV nsp1. The potential mechanisms of how the viral genome escape from the translation suppression by nsp1 should be discussed. Different models have been built to answer the question recently. Also, there is a rapid accumulation of structures of nsp1 and nsp1-ribosome complexes, which should be summarized, or at least properly cited. In addition, it would be good if the loss-of-function mutations of nsp1 are summarized in the review. Finally, more and more naturally occurring mutations of SARS-CoV-2 are arising now. It would substantially add to the value of the review if known mutations of SARS-CoV-2 nsp1 can be summarized and analyzed.

A minor point, “Matthias et al.” on line 207 should be “Thoms et al.” The last name of the referenced author is wrong.

Author Response

We thank reviewer’s comments/suggestions. We have modified the manuscript by using a track-change function. In addition to rearranging several sections of the manuscript, we added/modified several sentences, which are shown in blue.

Comment: Giving the ongoing pandemic and hence the interest of the community, one particular area that can be improved is a more detailed description of the specifics of SARS-CoV-2 nsp1, beyond what has been learned from the original SARS-CoV nsp1.

Our response: We modified two different sections of the manuscript to responses to the reviewer’s suggestion (lines 228-237 and lines 330-346).

Comment: The potential mechanisms of how the viral genome escape from the translation suppression by nsp1 should be discussed. Different models have been built to answer the question recently.

Our response: We provided description of the proposed models in lines 333-338.

Comment: Also, there is a rapid accumulation of structures of nsp1 and nsp1-ribosome complexes, which should be summarized, or at least properly cited.

Our response: We provided more explanation in lines 228-237.

Comment: It would substantially add to the value of the review if known mutations of SARS-CoV-2 nsp1 can be summarized and analyzed.

Our response: Section 9 of the original manuscript described naturally occurring nsp1 mutants. The revised manuscript describes SARS-CoV-2 variant carrying this naturally occurring nsp1 mutation in Perspective section (line 464-478).

Comment: A minor point, “Matthias et al.” on line 207 should be “Thoms et al.” The last name of the referenced author is wrong.

Our response: We have corrected this error (line 221).

Reviewer 2 Report

This review by Nakagawa and Makino focuses on the nsp1 protein SARS-CoV-2.  It is well written and explores a large body of literature on nsp1 biology. I very much like Table 1 as a resource to those interested in nsp1 or other SARS-CoV-2 proteins.

-Several odd statements in abstract. Unclear what degrading host mRNA means? That has a very specific mechanism for degradation but little is presented for concept in this phrase. “viral mRNAs evade this nsp1-induced”, unclear where the plural form of mRNAs comes from.

-A bit unclear why to discuss the cell free in vitro assays for nsp1 (lines 128-133) and if these are the basis for further discoveries discussed after. In addition, with all the discussion of gene regulation by promoter assays, nothing is addressed in the body of the paper about the localization of nsp1 in the cell and if it is getting into the nucleus to truly be regulating human genes outside of plasmid based or cell free systems.

- Line 217-219: “These structural based-insights into nsp1-40S ribosome interactions may facilitate rational design of antiviral drugs tar-geting the SARS-CoV-2-nsp1-ribosome interaction.” This is a high risk statement. The difficultly in targeting anything to do with nsp1 is the rapid drift and evolution of the sequence, especially targetable sites. Needs a statement regarding this issue and challenge of targeting. Right now no major pharma would invest in this as it is considered too high of risk.

-I do not understand why section 8 on accessory protein 5b is included, just seems out of place and detracts from focus of nsp1 or article.

Author Response

We thank reviewer’s comments/suggestions. We have modified the manuscript by using a track-change function. In addition to rearranging several sections of the manuscript, we added/modified several sentences, which are shown in blue.

Comment: Several odd statements in abstract. Unclear what degrading host mRNA means? That has a very specific mechanism for degradation but little is presented for concept in this phrase. “viral mRNAs evade this nsp1-induced”, unclear where the plural form of mRNAs comes from.

Our response: The abstract was modified; we used endonucleolytic RNA cleavage, instead of mRNA degradation. Several virus specific mRNAs are produced in CoV-infected cells; hence we think that the use of “viral mRNAs” is appropriate.

Comment: A bit unclear why to discuss the cell free in vitro assays for nsp1 (lines 128-133) and if these are the basis for further discoveries discussed after.

Our response: We modified the section to show an example for the usefulness of an in vitro assay (line 134-137).

Comment: In addition, with all the discussion of gene regulation by promoter assays, nothing is addressed in the body of the paper about the localization of nsp1 in the cell and if it is getting into the nucleus to truly be regulating human genes outside of plasmid based or cell free systems.

Our response: Subcellular localization of SARS-CoV nsp1, MERS-CoV nsp1 and TGEV nsp1 are described in lines 211-213, 363-366 and 446-447, respectively.

Comment: - Line 217-219: “These structural based-insights into nsp1-40S ribosome interactions may facilitate rational design of antiviral drugs targeting the SARS-CoV-2-nsp1-ribosome interaction.” This is a high risk statement. The difficultly in targeting anything to do with nsp1 is the rapid drift and evolution of the sequence, especially targetable sites. Needs a statement regarding this issue and challenge of targeting. Right now no major pharma would invest in this as it is considered too high of risk.

Our response: We provided our responses to the reviewer’s comment in Perspective section (lines 479-496).

Comment: -I do not understand why section 8 on accessory protein 5b is included, just seems out of place and detracts from focus of nsp1 or article.

Our response: We eliminated section 8 and provide a brief description of 5b in Perspective section (lines 497-506).

Reviewer 3 Report

OVERALL OPINION:

This review is focused on the host transcript/protein regulation of CoV Nsp1 and delves into the similarities and differences of pathogenic CoVs, mainly SARS-CoV and MERS-CoV,. The review achieves its goal, enabling the reader to gain a wider picture of the inhibition of transcription/translation by CoVs and its implication for pathogenicity. I lack information or perspective about SARS-CoV-2 that should be developed. The introduction is didactic and the information in figures and tables is reasonable well organized. Nevertheless, I have some comments that hopefully will improve the quality and usefulness of this review:

A major comment:

A review should not be a “copy-paste” exercise from published papers in PubMed, instead it should be a critical analysis and integrative work of the literature. The point 9 Emergence of SARS-CoV-2 strains carrying a deletion in nsp1 it is an example:

SARS-CoV-2 Nsp1 protein has 180aa, how come the authors comment about the deletions of residues 241-242-243 and its functional consequences? Although, this is a mistake that  has gone through the Journal (Journal of translational medicine 2020, 18, 329, doi:10.1186/s12967-020-02507-5) it should not be perpetuated in a “careless” Nsp1 review.

Minor points:

  1. I would be grateful if the authors could speculate a bit on the possibility of nsp1 as a druggable target in the “Perspective” section.
  2. I think the review will be easier to read by reorganizing some of the items. Information under bullet point 4.4 is a bit repetitive and can be distributed in points 3, 4.1, and 4.2. The same with bullet point 4.5 that could be put together with point 3.
  3. Figure 3 legend is too detailed. Most of it is already explained in the body of the review and results repetitive.
  4. Bullet point 9 may be dispensable in its current form. It contributes very little to understand either the biological function or importance of Nsp1 in this review. I suggest going more in depth on this item.
  5. In figure 1, the different CoVs are named either by the virus name or by the strain. The authors should use the same type of label for all the CoVs. In the figure legend, there is no information on the SARS-CoV-2 strain used to make the phylogenetic tree.
  6. Line 161: the authors state that Nsp1 of porcine epidemic diarrhea virus inhibits IkappaBalpha phosphorylation and promotes its degradation, leading to suppression of IFN production. However, inhibiting IkappaBalpha phosphorylation blocks its degradation and that is how NF-kappaB pathway is inhibited and IFN production suppressed. Please clarify.
  7. Lines 201 and 209: The mutated aminoacid at position 165 is a histidine, not a lysine. It is correct in line 255. Please correct
  8. Line 331: “in both expressed and infected cells” should be “in both transfected and infected cells”.

Author Response

We thank reviewer’s comments/suggestions. We have modified the manuscript by using a track-change function. In addition to rearranging several sections of the manuscript, we added/modified several sentences, which are shown in blue.

Major comment:

Comment: A review should not be a “copy-paste” exercise from published papers in PubMed, instead it should be a critical analysis and integrative work of the literature. The point 9 Emergence of SARS-CoV-2 strains carrying a deletion in nsp1 it is an example.

Our response: We removed description given in the point 9 of the original manuscript to Perspective sections and discussed the published data and their significance (line 464-478).

Comment: SARS-CoV-2 Nsp1 protein has 180aa, how come the authors comment about the deletions of residues 241-242-243 and its functional consequences? Although, this is a mistake that  has gone through the Journal (Journal of translational medicine 2020, 18, 329, doi:10.1186/s12967-020-02507-5) it should not be perpetuated in a “careless” Nsp1 review.

 Our response: We have corrected incorrect description of these amino acids (lines 470).

Minor points:

  1. I would be grateful if the authors could speculate a bit on the possibility of nsp1 as a druggable target in the “Perspective” section.

Our response: We discussed this in second paragraph of Perspective section.

  1. I think the review will be easier to read by reorganizing some of the items. Information under bullet point 4.4 is a bit repetitive and can be distributed in points 3, 4.1, and 4.2. The same with bullet point 4.5 that could be put together with point 3.

Our response: We combined 4.4 and 4.3 in the revised manuscript. We also moved descriptions given in 4.5 of the original manuscript to 3 of the revised manuscript.

  1. Figure 3 legend is too detailed. Most of it is already explained in the body of the review and results repetitive.

Our response: The revised manuscript now has shortened Fig. 3 legend.

  1. Bullet point 9 may be dispensable in its current form. It contributes very little to understand either the biological function or importance of Nsp1 in this review. I suggest going more in depth on this item.

Our response: We moved section 9 of the original manuscript to first paragraph of Perspective section and provided further discussion (line 464-478).

  1. In figure 1, the different CoVs are named either by the virus name or by the strain. The authors should use the same type of label for all the CoVs. In the figure legend, there is no information on the SARS-CoV-2 strain used to make the phylogenetic tree.

Our response: We modified Fig. 1 and provided accession number of SARS-CoV-2 strain that was used in the phylogenetic tree.

  1. Line 161: the authors state that Nsp1 of porcine epidemic diarrhea virus inhibits IkappaBalpha phosphorylation and promotes its degradation, leading to suppression of IFN production. However, inhibiting IkappaBalpha phosphorylation blocks its degradation and that is how NF-kappaB pathway is inhibited and IFN production suppressed. Please clarify.

Our response: We thank the reviewer for pointing out our incorrect statement. We altered the sentence (172-174).

  1. Lines 201 and 209: The mutated amino acid at position 165 is a histidine, not a lysine. It is correct in line 255. Please correct.

Our response: We thank the reviewer for pointing out this error. We corrected the error (lines 213, 215 and 223)

  1. Line 331: “in both expressed and infected cells” should be “in both transfected and infected cells”.

Our response: We altered the sentence to clarify the meaning (lines 351-352).

Round 2

Reviewer 3 Report

The authors had improved the manuscript.